# Transcranial Direct Current Stimulation Decreases P3 Amplitude and Inherent Delta Activity during a Waiting Impulsivity Paradigm: Crossover Study

**DOI:** 10.3390/brainsci14020168

**Published:** 2024-02-07

**Authors:** Augusto J. Mendes, Santiago Galdo-Álvarez, Alberto Lema, Sandra Carvalho, Jorge Leite

**Affiliations:** 1Psychological Neuroscience Laboratory, CIPsi, School of Psychology, University of Minho, Campus de Gualtar, 4704-553 Braga, Portugal; augustojmmendes@gmail.com (A.J.M.); alberto.lemac@gmail.com (A.L.); 2Laboratory of Neuroimaging of Aging (LANVIE), University of Geneva, 1205 Geneva, Switzerland; 3Geneva Memory Center, Department of Rehabilitation and Geriatrics, Geneva University Hospitals, 1205 Geneva, Switzerland; 4Laboratorio de Neurociencia Cognitiva, Departamento de Psicoloxía Clínica e Psicobioloxía, Facultade de Psicoloxía, Universidade de Santiago de Compostela, 1205 Galicia, Spain; santiago.galdo@usc.es; 5Department of Education and Psychology, William James Center for Research (WJCR), University of Aveiro, Campus Universitário de Santiago, 3810-193 Aveiro, Portugal; 6CINTESIS@RISE, Center for Health Technology and Services Research at the Associate Laboratory RISE—Health Research Network, Department of Education and Psychology, University of Aveiro, 3810-193 Aveiro, Portugal; 7CINTESIS@RISE, CINTESIS.UPT, Portucalense University, 4200-072 Porto, Portugal; jorgel@upt.pt

**Keywords:** waiting impulsivity, premature responses, tDCS, rIFG, P3, delta, theta

## Abstract

The inability to wait for a target before initiating an action (i.e., waiting impulsivity) is one of the main features of addictive behaviors. Current interventions for addiction, such as transcranial Direct Current Stimulation (tDCS), have been suggested to improve this inability. Nonetheless, the effects of tDCS on waiting impulsivity and underlying electrophysiological (EEG) markers are still not clear. Therefore, this study aimed to evaluate the effects of neuromodulation over the right inferior frontal gyrus (rIFG) on the behavior and EEG markers of reward anticipation (i.e., cue and target-P3 and underlying delta/theta power) during a premature responding task. For that, forty healthy subjects participated in two experimental sessions, where they received active and sham tDCS over the rIFG combined with EEG recording during the task. To evaluate transfer effects, participants also performed two control tasks to assess delay discounting and motor inhibition. The active tDCS decreased the cue-P3 and target-P3 amplitudes, as well as delta power during target-P3. While no tDCS effects were found for motor inhibition, active tDCS increased the discounting of future rewards when compared to sham. These findings suggest a tDCS-induced modulation of the P3 component and underlying oscillatory activity during waiting impulsivity and the discounting of future rewards.

## 1. Introduction

Transcranial Direct Current Stimulation (tDCS) over the prefrontal areas has already shown promising effects for addiction [1], craving [2], and reward responsiveness [3]. Studies using tDCS to target inhibitory processes have been targeting several cortical regions, such as the right inferior frontal gyrus (rIFG), related to response inhibition [4], or the dorsolateral prefrontal cortex (DLPFC), in order to enhance delay discounting [5]. Although both processes are framed within impulsivity (i.e., stopping vs. waiting), they differ in their cognitive procedures and neuronal networks. For instance, waiting impulsivity relies on the top-down regulation of the prefrontal cortex, involving subcortical structures such as the ventral striatum, amygdala, and hippocampus [6]; while stopping requires interactions between the rIFG and the pre-supplementary motor area (SMA), with the dorsal striatum and the subthalamic nucleus (STN) [7].

Waiting impulsivity can be further divided into two main dimensions, namely the impulsive action measured by premature responses and the impulsive choice assessed in delay discounting tasks. This distinction highlights the role of the impulsive action as a ‘cold’ process, less prone to affective/emotional influences, whilst the impulsive choice is thought to be a ‘hot’ process relying on reward processing [8]. Premature responses have also been associated with proactive stopping evaluated in Go/No-Go (GNG) paradigms, given that inhibitory processes may be prior to the response selection [9]. The GNG is a task composed of two types of trials, namely a frequent “Go” trial and an infrequent “No-Go”. Participants are required to press a button during the “Go”; however, in “No-Go” trials, they are required to inhibit their response. On the other hand, reactive inhibition paradigms, such as the Stop Signal Reaction Time task (SSRTT), have not been associated with premature responses, given that the inhibitory processes act after a response has started [10]. This is because the SSRTT incorporates a “stop” trial, during which participants are required to suppress a response that was initiated during a “Go” trial in response to a “stop” stimulus that is presented immediately after the “Go” stimulus.

Furthermore, attentional and inhibitory processes have been studied using event-related potentials (ERPs), mainly through the P3 component. P3 is characterized by a positive prominent wave at centro-parietal sites with a peak between 250 and 600 ms after the presentation of a task-relevant stimulus [11]. P3 can also be elicited when rewards/punishments are anticipated. The anticipatory or cue-P3 is elicited after the cue onset, and it has been interpreted as the motivated attention to a subsequent task-relevant stimulus [12]. The consummatory or target-P3 is the actual response toward the motivational stimulus predicted by the cue [13]. These components have been tested in the Monetary Incentive Delay (MID) task, in which participants are rewarded or punished according to the response time towards a cued target stimulus [13]. The cue indicates the type of trial (i.e., win or loss), followed by a target (with a jittered interval) and immediate feedback about performance. Thus, the cue-P3 is elicited following the cue-onset that predicts the incoming target and the type of feedback (i.e., reward/punishment), which has been suggested to represent the motivated attention towards the impending relevant-stimulus (i.e., target) [14]. The target-P3 is elicited following the onset of a stimulus implied in the subsequent reward or punishment process. Thus, target-P3 amplitude increases after cues that predict gains or losses [13]. However, the specific effects in terms of the valence of reward in cue-P3 are not clear [13,15,16,17]. Specifically, cue-P3 is greater in trials predicting rewards [13], losses (in schizophrenia subjects) [17], or both [15,16,17] in comparison with neutral trials. Moreover, cue-P3 amplitude was increased in win trials compared to loss trials [12,15]. Overall, both P3 components are enhanced in reward-laden trials, suggesting attentional allocation to motivational stimuli [16].

Assuming that ERPs are representations of specific brain activations in the time domain, they co-occur with the synchronization of oscillatory activity in the time–frequency domain, i.e., the event-related oscillations (ERO) [18]. Regarding the EROs, the P3 elicited during the oddball paradigm is accompanied by a transient increase in delta (0.5–4 Hz) and theta (4–7 Hz) power at the same latency and scalp distribution of P3 [19]. These findings suggest that both oscillatory frequencies might represent the mechanism of the generation of the P3 component [20]. Additionally, an increase in parietal delta power after a reward-laden stimulus has been shown, suggesting that the association between P3 and delta might also occur during the anticipation of rewards [21]. Nevertheless, this concurrent activity between both EEG markers in parietal regions has not been tested yet in MID tasks or other reward-processing paradigms.

In a recent meta-analysis about the modulatory effects of tDCS in the P3 component during cognitive tasks, we were able to show an increase in the parietal P3 in attentional and working memory tasks after tDCS to the frontal cortex [22]. The effects of tDCS in terms of P3 amplitude elicited during response inhibition were mixed: some studies showed an increase in P3 amplitude [23], while others a decrease [24,25].

Even though previous tDCS studies have targeted addictive/impulsive behaviors, the available knowledge about neural correlates of impulsivity highlights the importance of understanding how tDCS impacts cognition but also the underlying neuronal activity. For instance, a recent study guided tDCS through an online analysis of EEG data collected during the performance of a cognitive task [26]. This closed-loop methodology aimed to augment the effects of tDCS but also relied on the detection of specific neuro-markers of cognition. Therefore, the understanding of the effect of tDCS on cognition, as well as the understanding of the effects of tDCS in EEG markers related to cognition, is of the utmost importance for the optimization of the effects. In this sense, the aim of the current study was to assess the effects of tDCS over rIFG in (i) premature responses and (ii) anticipatory and consummatory ERPs and EROs related to waiting impulsivity. For this purpose, we developed a computerized task to test the anticipatory and consummatory neural response towards a salient stimulus involved in reward-seeking, which was tailored to individual performance. Considering the essential role of rIFG in response inhibition [27], we hypothesized that active tDCS over the rIFG would lead to a reduction in the number of premature responses. Moreover, this reduction in the number of premature responses may also show a transfer effect for delay discounting and inhibitory control. Likewise, the behavioral modulation is expected to be combined with an increase in P3 amplitude and consequently delta/theta power.

## 2. Materials and Methods

### 2.1. Participants

A total of 40 healthy volunteers (31 females; mean age: 23.2 ± 3.52) participated in the study. All the participants were right-handed (Edinburgh Handedness Inventory > 40) and without a recent history of neurological or psychiatric disorders. Prior to their participation in the experiment, volunteers were assessed using the Beck Depression Inventory-II [28], Alcohol Use Disorders Identification Test [29], and Drug Use Disorders Identification Test [30] to ensure the absence of depressive symptomatology and the consumption of alcohol/drugs. Moreover, participants who reported medication or psychotropic drug consumption during the 4 weeks before the study were not included. All participants gave their written informed consent preceding their enrollment. The study was in accordance with the Declaration of Helsinki and was approved by the local ethics committee (CEICVS 127/2019).

### 2.2. General Procedure

The study comprised two sessions, one with active and another with sham tDCS, which were administered randomly to each participant in a counterbalanced order (see Appendix A). Both sessions followed similar procedures, except for the screening, informed consent, and self-report questionnaires, which were performed only in the first session. In each session, participants performed the Cued Premature Response Task (CPRT) (Figure 1) during tDCS. EEG data were collected during the entire performance of the CPRT. Then, participants performed the SSRTT [31] and the Monetary Choice Questionnaire-27 (MCQ-27) [32] in a counterbalanced order to assess potential far-transfer effects for delay discounting and inhibitory control. At the end of the session, participants filled a blinding questionnaire about the tDCS condition and the side effects of tDCS using a Visual Analog Scale from 0 to 10 (see Appendix A).

### 2.3. Cued Premature Response Task

The experimental task was developed to assess premature responding during monetary reinforcement and punishment. For that, CPRT was adapted from the 4-Choice Serial Reaction Time Task [33] and the MID [13]. Participants were instructed to press the middle button of the E-Prime Chronos response box to start a trial and to release it as fast as possible when the target was displayed on the screen. The target was always preceded by a cue, which informed the participant that the target was about to be displayed. The cue and the target were always displayed with a random onset to minimize expectancy. Specifically, the interval between the trial onset and the cue ranged from 1250 to 1750 ms, and the interval between the cue and the target ranged from 500 to 2500 ms (Figure 1A). However, in the baseline block, these intervals were fixed at 1000 ms.

Participants were instructed to release the button after target, as fast as possible, thus favoring speedier responses instead of more accurate, albeit slower, responses. Participant’s responses were rewarded with virtual money if their responses were faster, punished if their responses were slower, or neither rewarded nor punished if they responded before target onset (i.e., premature responses; Figure 1B) [33]. The task comprised one training block with 20 trials and one test block with 180 trials in total. Participants started to perform the test block task after three minutes of the onset of tDCS, and the total duration of the task was approximately 15 min. The behavioral outcomes analyzed were the number of premature responses, the monetary gain/loss, and the RT average (details in Appendix A).

### 2.4. Control Assignments

#### 2.4.1. Stop-Signal Reaction Time Task

The SSRTT is designed to assess inhibitory control abilities, and the version used in the study followed the latest guidelines for the appropriate structure of the task [34]. For this purpose, participants were instructed to respond according to the orientation of an arrow. Specifically, if an arrow was pointing to the left, the participant should press the left button (“Z”); if the arrow was pointing to the right, the participant should press the right button (“M”). Moreover, for stop trials, a red frame around the arrow (i.e., stop-signal) was shown on the screen, which informed participants to withhold any response. The stop signal appeared after the arrow was displayed, with a delay adjusted for each participant following a staircase-tracking algorithm [31]. The Stop-Signal Delay (SSD) was set to 250 ms at the beginning of each block, and it was adjusted after an unsuccessful inhibition (−25 ms) and a successful inhibition (+25 ms). The maximum SSD was 400 ms, and the minimum was 0 ms. The goal of this adjustment was to guarantee a *p*(response|stop-signal) of 0.5. Therefore, participants were instructed to be as fast and accurate as possible, even though they should not wait for the stop signal to control for the speed-accuracy tradeoff.

The task started with a training block of 24 trials, followed by four experimental blocks with 64 trials each. Each block comprised 75% go trials (i.e., 48 trials) and 25% stop trials (i.e., 16 trials). However, the first six trials of a block were always a go trial. The task had a total duration of approximately 8 min. The outliers were identified following the lenient criteria from Congdon and colleagues [35]. The outcomes included in the statistical analysis were the accuracy and RT of go trials, *p*(respond|signal), SSD, and the stop-signal reaction time (SSRT).

#### 2.4.2. 27-Item Monetary Choice Questionnaires

The MCQ-27 was administered to evaluate the discount rating (i.e., if the participant preferred smaller but immediate rewards instead of larger but delayed rewards). The questionnaire was composed of 27 questions with two possible answers, namely smaller and immediate reward or a larger, albeit delayed, reward [32]. The outcomes of the questionnaire were the overall, small, medium, and large k. The overall k represented the steepness of the discounting for all the monetary values (i.e., taking into consideration all the items from the questionnaire), while the small, medium, and large k were specific to the corresponding amounts (i.e., taking into consideration nine items from the questionnaire per amount). The discount rates (i.e., k) were calculated in an Excel-based spreadsheet scoring tool [36].

### 2.5. Transcranial Direct Current Stimulation

Participants received both active and sham tDCS in distinct sessions through a Starstim R20 (Neuroelectrics, Barcelona, Spain). For the active tDCS condition, an electric current of 2 mA of intensity was applied for 20 min (with 15 s of ramp up and ramp down) while the participant performed the CPRT. The sham procedure was similar to the active stimulation but with only a 45 s duration (with 15 s of ramp up and ramp down). 25 cm^2^ round saline-soaked electrode sponges (~radius of 3 cm, current density: 0.08 mA/cm²) were placed over F8 (anode) and posterior to the left mastoid (cathode) [37]. This tDCS montage was chosen because, according to computer modeling, the current densities are higher over the right inferior frontal areas [37]. Moreover, the placement of the cathode electrode on the left mastoid may prevent potential dual effects from other relevant brain areas [38].

### 2.6. Electrophysiological Acquisition and Data Analysis

Online EEG data were collected with a Starstim R20 (Neuroelectrics, Barcelona, Spain) using 18 scalps electrodes and one earlobe electrode. Electrophysiological data were preprocessed offline and analyzed using EEGLAB [39]. Data were sampled at a rate of 500 Hz and FIR filtered with a bandpass between 0.5 and 40 Hz. The DC offset was removed as the line noise using a notch filter (i.e., 50 Hz). The artifacts in the continuous data were corrected, and noisy channels were removed using the Artifact Subspace Reconstruction in the clean_rawdata function. The parameters for the identification of noisy channels were the following: flatline with a maximum duration of 5 s and correlation between channels below 0.7. EEG data were re-referenced to the average but without the pre-identified noisy channels, and the rejected channels were interpolated using the spherical spline method [40]. An average of 1.85 channels were rejected (SD = 1.09) in datasets from the active sessions and 1.78 (SD = 0.92) from the sham ones.

The continuous data was segmented in epochs with a total length of four seconds (i.e., 2000 ms prior and post-stimulus onset) centered in the target and cue. The epochs containing premature responses in the 1000 ms post-cue onset were excluded. Moreover, due to the time window chosen for the cue-P3, only epochs with a cue–target interval of at least 800 ms were selected. The epochs in which the EEG signal surpassed ±150 µV at non-frontal electrodes were rejected. All the epochs were visually inspected and manually removed in the plot window if artifacts were present. At last, an independent component analysis (ICA) was performed to detect and remove artifacts using the ICLabel [41] (for outliers’ information see Appendix A).

#### 2.6.1. Event-Related Potentials

The ERPs analyzed were the cue-P3 and target-P3 in the Pz electrode. The target-P3 and cue-P3 epochs were baselined to the 200 ms pre-stimulus interval. The time windows selected for each ERP were based on a previous study [13], and the data were averaged following these time windows: the target-P3 was the average amplitude between 250 and 450 ms, and the cue-P3 was between 350 and 600 ms.

#### 2.6.2. Event-Related Oscillations

The ERO power was analyzed using the Event-Related Spectral Perturbation (ERSP) in EEGLAB function newtimef() [39]. An additional analysis was performed regarding the ERO phase, specifically the magnitude of Inter-Trial Phase Coherence (ITPC) (see Appendix A). For that, a time–frequency decomposition using three-cycle Morlet wavelets with a frequency resolution of 0.25 Hz and temporal resolution of 8 ms was applied. The analyzed frequencies ranged from 1.5 Hz to 20 Hz for the cue and target epochs. The ERSP baseline normalization followed an unbiased single-trial baseline correction (i.e., full-epoch length single-trial corrections) to minimize sensitivity to noisy trials [42]. Therefore, at first, the average activity from the full-epoch length (i.e., µV) was subtracted from each epoch. Subsequently, the spectral power was averaged considering all the trials according to the baseline window (i.e., 1000 ms pre-cue or target). Finally, following the additive model [42,43], each epoch was normalized by the previously calculated power spectral average. The ERSP was averaged for delta (1.5–4 Hz) and theta (4–7 Hz) bands [20] following the same electrode (i.e., Pz) and time-windows (i.e., target-P3: 250–450 ms; cue-P3: 350–600 ms) from the ERP analysis [13].

### 2.7. Statistical Analysis

The analysis focused on the difference between the active and sham stimulation conditions. Therefore, paired *t*-tests were performed when the difference between both conditions followed the normal distribution, as assessed by the Shapiro–Wilk test. If there was no normal distribution, Wilcoxon signed rank test on paired samples was performed instead. Holm–Bonferroni (BH) correction was also performed in each section of the statistical analysis for multiple comparisons. At last, to probe the association between the number of premature responses and the average release time on the Baseline block, Pearson or Spearman correlations (depending on the normality in both variables) were performed to evaluate the association between the reward/punishment system in the number of premature responses. The statistical analysis was performed in R, version 4.0.3 [44].

## 3. Results

### 3.1. Behavioral Analysis

#### 3.1.1. Cued Premature Response Task

No significant effects of tDCS were observed in premature responses (*t*(35) = −0.79, *p* = 0.438), the monetary amount earned (*t*(37) = 0.78, *p* = 0.438), and release time (*t*(37) = −0.87, *p* = 0.438) (Table 1). However, the number of premature responses and the average release time for the baseline block were significantly correlated during active (R = −0.33, *p* = 0.0045) and sham sessions (R = −0.34, *p* = 0.035), as revealed in Spearman correlations, suggesting that the reward/punishment system was associated with the posterior number of premature responses (Figure 2).

#### 3.1.2. Stop Signal Reaction Time Task

The *t*-tests did not reveal any significant effect in terms of Go trials accuracy (*t*(30) = −2.11, *p* = 0.215), Go trials response time (*t*(30) = 1.09, *p* = 0.707), *p*(respond|signal) (*t*(30) = −0.38, *p* = 0.71), SSD (*t*(30) = −0.52, *p* = 0.71), or SSRT (*t*(30) = 0.73, *p* = 0.71) (Table 1).

#### 3.1.3. 27-Item Monetary Choice Questionnaires

There was a significant effect in the Small k (V(39) = 199, *p* = 0.016), suggesting a higher k in small amounts of money for the active tDCS session compared to sham. However, there were no other differences due to tDCS being found for the 27-MCQ, namely Overall k (V(39) = 344, *p* = 0.12), Medium k (V(39) = 134, *p* = 0.12), and Large k (V(39) = 223, *p* = 0.12) (Table 1).

### 3.2. EEG Analysis

#### 3.2.1. Event-Related Potentials

The Wilcoxon signed rank test revealed a significant difference between active and sham sessions for the target-P3 amplitude (V(32) = 106, *p* = 0.003) and the cue-P3 (V(32) = 142, *p* = 0.036). The target and cue-P3 amplitude was significantly lower in the active session when compared to sham (Figure 3B and Figure 4B; Table 2).

#### 3.2.2. Event-Related Oscillations

During the target-P3 time window, the event-related synchronization in the delta band during the target-P3 was significantly higher during sham compared to active tDCS (*t*(32) = −2.29, *p* = 0.03) (Figure 3C and Figure 4C). However, no differences were found for theta band power (*t*(32) = −0.66, *p* = 0.805). Regarding the cue-P3 time window, *t*-tests did not reveal any significant effect in terms of ERO, namely for delta (*t*(32) = −0.24, *p* = 0.847) and theta bands (*t*(32) = −0.19, *p* = 0.847) (Table 2).

## 4. Discussion

The current study shows that tDCS over the right inferior frontal gyrus can modulate the P3 component and underlying oscillatory activity during a waiting impulsivity task (CPRT). Namely, active tDCS induced a decrease in target and cue-P3 amplitudes. Moreover, the reduction in the target-P3 amplitude during active stimulation was combined with a simultaneous reduction in terms of delta power for the same time window. Regarding behavioral analysis, there was a significantly higher k in the small amounts condition after active tDCS in comparison with sham, suggesting a preference for small immediate rewards rather than bigger, delayed ones during an active tDCS session. However, no modulatory effects of tDCS over rIFG were found in terms of waiting impulsivity and inhibitory control measures (i.e., CPRT and SSRTT, respectively).

### 4.1. Electrophysiological Correlates

In the present study, anodal tDCS over the rIFG decreased the target-P3 amplitude and underlying oscillatory activity (namely delta power) during a waiting impulsivity task. However, there is a growing body of evidence suggesting that anodal tDCS over frontal areas can increase the P3 amplitude in tasks involving attentional and working memory processes [22]. However, the effects of tDCS over frontal regions on the P3 amplitude during inhibitory control paradigms are mixed. While some studies report a decreased [24,25] P3 amplitude following tDCS, others report an increased P3 amplitude following tDCS [23]. Thus, the differential effects of tDCS on P3 may be related to the functional role of each cognitive task and its underlying neuronal substrates [22].

Furthermore, these findings underpin the relationship between P3 and the delta/theta power at the same time window observed in several cognitive tasks [20,45]. A recent study showed an enhancement in the P3 amplitude during a visual oddball paradigm after the entrainment of delta/theta frequency bands through the application of transcranial Alternating Current Stimulation (tACS) [46]. However, theta activity was not modulated concurrently to both the cue and target-P3 amplitudes. Surprisingly, it was delta power that was modulated. This may show a potential inter-dependency between both electrophysiological markers and their importance during impulsive behaviors. For instance, a study showed decreased impulsive eating behavior in rats through a closed-loop system that triggered responsive neurostimulation in the nucleus accumbens every time delta activity was excessively increased during reward anticipation [47]. Likewise, delta power and P3 amplitude in the parietal region are also enhanced during the anticipation of rewards when compared to neutral trials [21]. This is of particular interest because a positive correlation between the cue-P3 amplitude and the activity in the ventral striatum (including the nucleus accumbens) has been shown before [12]. It is important to highlight that the ventral striatum is a core structure for reward processing and impulsive choice [12]. Similarly, the neuronal activity observed on the left ventral striatum activity during an inhibitory control task was negatively correlated with the rIFG [48]. Thus, this might suggest that tDCS over the rIFG might not only reduce the P3 amplitude and delta power but also decrease ventral striatum activity.

In addition, the anticipatory (i.e., cue) and consummatory (i.e., target) P3 in tasks with monetary incentives are strongly related to reward processing. The target-P3 amplitude is greater when preceded by cues that predict either a winning or loss of monetary compensation, thus suggesting the involvement of the P3 component in reward and punishment processing [13]. The involvement of the cue-P3 in this reward and punishment processing is more mixed, as there are studies suggesting a cue-P3 enhancement after reward cues [12,13,15], loss cues [17], or both [16]. For ERO, cues that predict rewards elicited an enhancement in delta power in parietal areas (and cue-P3 amplitude) when compared to neutral cues [21]. Although the previous studies did not show a significant relation between cue-delta activity and delay discounting, a recent study showed that the increase in evoked delta during a delay discounting task was associated with the choice for larger, albeit delayed, rewards [49]. Therefore, the decrease in P3 amplitude and delta activity might indicate a modulation in the impulsive choice identified in our delay discounting results (but not found in CPRT and SSRTT).

### 4.2. Behavioral Outcomes

tDCS over rIFG did not impact the CPRT outcomes, namely the number of premature responses, release time, and total earned money. These findings suggest that, although previous studies suggested the involvement of the rIFG in inhibitory control [24,27], the rIFG might not be critically involved in waiting impulsivity [9]. Specifically, we expected an increase in the tonic inhibitory process, which, in turn, would result in fewer premature responses. Our results did not support this hypothesis. Nonetheless, several reasons might be pointed out to explain the lack of tDCS effects in waiting impulsivity. First, tDCS over rIFG might show a greater effect in reactive inhibition than on tonic inhibitory response involved in premature responses. Indeed, a recent meta-analysis exploring the effects of tDCS in both inhibitory processes showed a significantly larger effect size in reactive inhibition (e.g., SSRTT) than in tonic inhibition (e.g., GNG task) [4]. Therefore, a smaller effect of tDCS over the rIFG could be expected, due to the association between proactive stopping and premature responses [9]. Additionally, the rIFG neural circuits involved in both reactive and proactive inhibition follow different pathways. An indirect pathway has been related to proactive inhibition, in which the rIFG connects with the globus pallidus through the dorsal striatum, while reactive inhibition is related to a hyperdirect pathway from the rIFG and pre-SMA to STN by-passing the striatum [50]. Moreover, the increase in terms of premature responses was associated with lower connectivity within structures relevant to motor inhibition, such as the STN and ventral striatum [10]. Therefore, differences in neural pathways might influence how tDCS affects the rIFG based on the network-dependent activity related to the CPRT [51]. This is supported by the absence of transfer effects from the waiting impulsivity task to the motor inhibition performance evaluated by the SSRTT. Nonetheless, this hypothesis is not in line with the previous literature, given that several studies targeting the same area showed an enhancement in proactive inhibitory processes [24,38,52,53,54].

Another explanation is that tDCS might increase the proactive inhibition but without any consequence in terms of premature responding. This dissociation was already observed in the literature [10,55]. Waiting and stopping have been suggested to represent distinct constructs within impulsivity [6]. They rely on different cortico-striatal connections between the DLPFC and ventral striatum for waiting processes and between the IFG and dorsal striatum for stopping [7,56]. Furthermore, the differences between the reward and the punishment systems might undermine our ability to make any conclusions about the effects of tDCS on premature responses or other outcomes from CPRT as suggested by the Pearson correlation. Specifically, when it was harder to win money, participants incurred more premature responses. Furthermore, as the baseline block was performed at the beginning of each section, the system of reward/punishment was also updated in each session (see Section 4.3).

Moreover, the preference for immediate and smaller rewards observed in the MCQ-27 might be explained by the activation of concurrent neuronal circuitries between waiting impulsive action and delay discounting [6]. This is of particular interest given that both processes depend on the ventral striatum, even though they share different pathways. Specifically, waiting impulsivity relies on the connectivity between the STN with the ventral striatum and subgenual cingulate cortex [10]. Increased magnitudes of delayed rewards were associated with the activation of mesolimbic pathways through the ventral striatum, medial prefrontal cortex, and posterior cingulate cortex [57]. In line with this, studies have shown an effect of tDCS over the DLPFC in the dopamine release in the ventral striatum [58,59], which might explain the transfer effect of tDCS to the delay discounting assessment. Similarly, a neuroimaging study showed that the activity observed in the rIFG was negatively correlated with the activity found in the left ventral striatum [48]. Therefore, the application of anodal stimulation over rIFG might result in a lower activation in the ventral striatum and consequently increase the k, as assessed by the MCQ-27. Nonetheless, to the best of our knowledge, this study was the first to test the effect of tDCS over rIFG in delay discounting.

In general, the tDCS transfer effects were only observed in terms of impulsive choice (i.e., delay discounting) that partially shares neuronal circuits with waiting impulsive action [6]. Therefore, the modulation of the neuronal circuits related to waiting impulsivity is in line with the tDCS model of the network activity-dependent model [51]. On the other hand, the lack of transfer effect in the inhibitory control task (i.e., SSRTT) might suggest the dissociation between waiting and stopping impulsivity [60] or between impulsive choice and action [8] (see Section 4.3 Limitations and Future Directions).

### 4.3. Limitations and Future Directions

The reward/punishment system was estimated for each session and not for each participant, which led to some differences in the mean RT and SD between sessions. This shift might misinterpret the effect of tDCS over the rIFG in the number of premature responses because the urge to prematurely release the button might be influenced by the reward/punishment system. Furthermore, this limitation might also influence the SSRTT because the differential behavioral training (i.e., due to the distinct reward/punishment system) might result in distinct plastic changes induced by tDCS [51]. This limitation might have a strong impact in terms of behavioral performance and therefore “mask” a potential effect of tDCS in the CPRT outcomes. Additionally, the tDCS effect observed on P3 amplitude and evoked-delta power may not be robust enough to lead to behavioral changes. For instance, during WM paradigms, Cespón and colleagues [61] showed that the increase of P3 amplitude after frontal tDCS was correlated with behavioral gains, while Breitling and colleagues [62] only found an enhancement in P3 amplitude by frontal tDCS, without any behavioral effects. Likewise, similar findings of non-overlapping effects on markers and behavior were shown during functional MRI, in which significant tDCS-induced changes in the BOLD signal were not accompanied by a significant modulation of behavior [63]. Therefore, a larger sample size should be preferred to evaluate the neuromodulatory effects in CPRT, as well as an individualized reward/punishment system per participant (and not per session).

Moreover, the interdependency between impulsive subtypes is still not clear within premature response paradigms [9]. In fact, most of the evidence was observed from animal studies and does not always match the one found in human studies [6]. This highlights the importance of P3 as a potential surrogate marker for the cognitive processing of impulsivity, which can be used for several clinical conditions, such as alcohol use disorder [64] and attention-deficit/hyperactivity disorder (ADHD) [65]. Therefore, tDCS and EEG studies are important to understand the underlying neural circuitries, as well as for the development of available interventions related to several clinical conditions [1]. Nevertheless, these studies must enhance their methods in analyzing the EEG data, reducing the impact of artifacts that can interfere with the data. In this particular study, we opted for a previously utilized method, but it is important for future studies to explore alternative approaches to mitigate the potential impact on the data.

In addition, applying tDCS to other relevant cortical areas should be addressed in future studies. For instance, tDCS studies aiming at the DLPFC have shown modulatory effects in several processes of impulsivity such as delay discounting abilities [66]. Furthermore, even though the computer modeling of electrical current densities suggests that most of the induced current is over the rIFG, other regions such as the DLPFC may also be stimulated. Thus, future studies should use a premature response paradigm to ascertain the relationship between both impulsive sub-processes, as well as to understand specific contributions from different regions of the task-related network, such as the DLPFC. On the other hand, other transcranial electrical stimulation techniques, such as tACS [46], or the application of closed-loop systems [26] in impulsive processes should be addressed in the future, given the strengthening of the association between P3 and oscillatory activity suggested by this study. However, although tDCS decreased both the P3 amplitude and delta power after the target, this was not observed after the cue. This finding might raise some questions about the association between the cue-P3 and delta/theta power in the time window suggested by Broyd and colleagues [13]. Therefore, the cue-P3 should be re-examined according to its functional role and the related oscillatory power during impulsive paradigms.

Finally, in the current study, the cue did not predict the winning or loss of money as in the studies previously mentioned [12,13,15,16], given that the reward or punishment could occur in each trial depending on the subject’s performance (i.e., the only way of not winning/losing virtual money was the premature response). The difference between positive and negative reinforcement should be evaluated in the future to fully understand the dynamics of P3/delta and waiting impulsivity/reward processing.

### 4.4. General Discussion

P3 amplitude has been shown to be significantly enhanced during reward anticipation [13,16]. In particular, P3 elicited during reward cues is positively correlated with neuronal activation in the ventral striatum evaluated by neuroimaging studies [12]. Likewise, activity in the ventral striatum was found to be negatively correlated with activity detected on the rIFG during a response inhibition paradigm [48]. These cortico-striatal dynamics are of particular interest because lower k was associated with larger activity in the ventral striatum [57]. This is in accordance with our EEG and behavioral results, given that tDCS over rIFG decreased the P3 amplitude and increased the choice for immediate rewards, which, in turn, have been associated with reduced activity in the ventral striatum [12,57]. Therefore, there is a possibility that the cortico-striatal interactions might explain the modulation of P3 and small k after tDCS over rIFG.

## 5. Conclusions

Overall, the current study suggests the decrease in anticipatory and consummatory P3 amplitude and underlying oscillatory activity (i.e., a decrease of delta power during target-P3) after tDCS over the rIFG. On the other hand, these variations were not accompanied by changes in terms of behavioral outcomes during waiting impulsivity, although a difference in delay discounting ability was detected between active and sham tDCS. These modulatory effects of tDCS are of particular interest due to the association between P3, delta power, and reward processing in patients with neuropsychiatric disorders. Moreover, these findings suggest the usefulness of studying tDCS-induced effects on ERPs and EROs in impulsive disorders, such as addiction or ADHD, as surrogate markers of cognitive processes.

## Figures and Tables

**Figure 1 brainsci-14-00168-f001:**
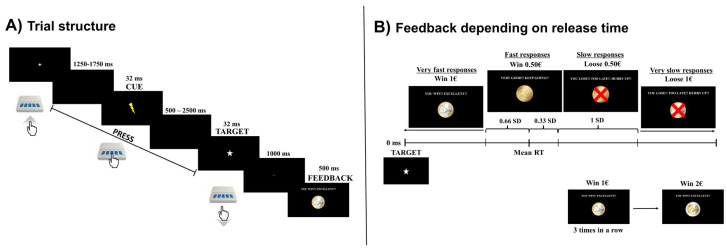
Overview of the experimental task (**A**) and the individualized reinforcement/punishment feedback (**B**).

**Figure 2 brainsci-14-00168-f002:**
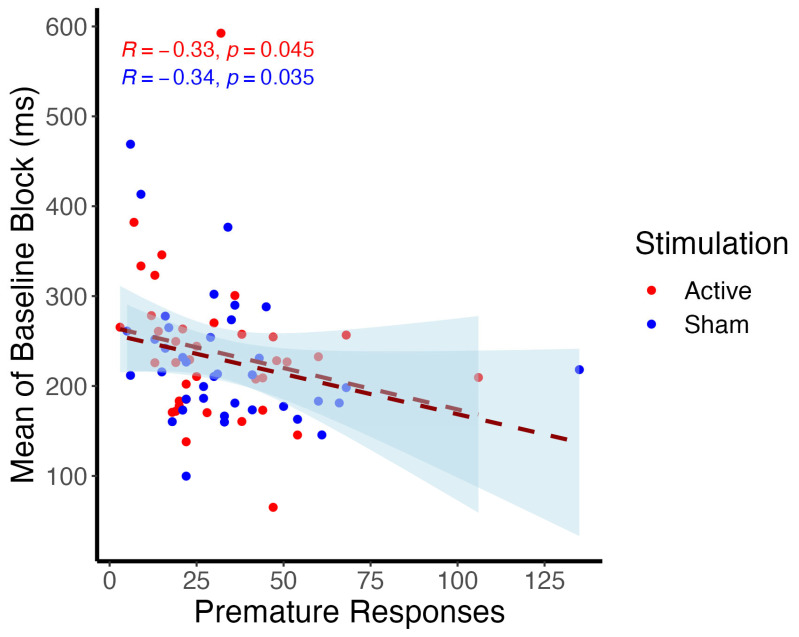
Correlation between the number of premature responses and the average release time in the baseline block.

**Figure 3 brainsci-14-00168-f003:**
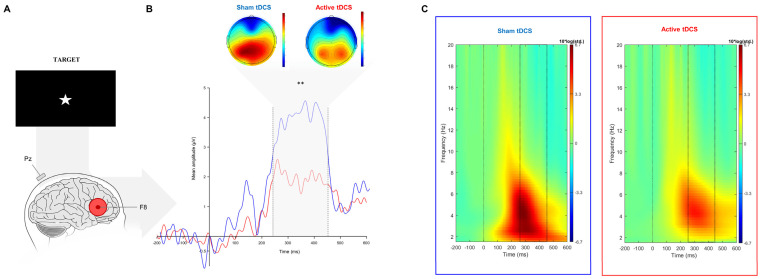
(**A**) Grand average event-related target-P3 at the Pz electrode (**B**) with topographical maps in the time-window of interest (represented in the gray area and dashed lines: 250–450 ms) and ERO results at Pz electrode (**C**) between both tDCS conditions. ** *p* < 0.01.

**Figure 4 brainsci-14-00168-f004:**
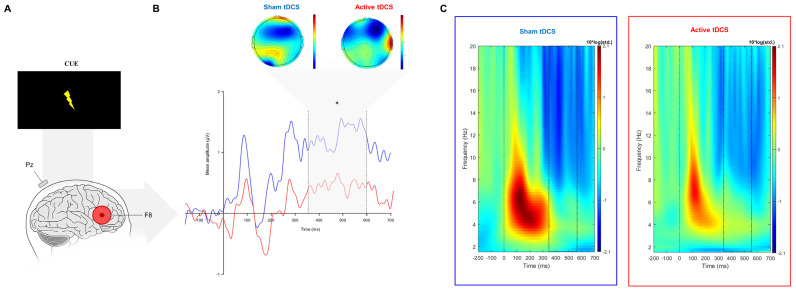
(**A**) Grand average event-related cue-P3 at Pz electrode (**B**) with topographical maps in the time-window of interest (represented in the gray area and dashed lines: 350–600 ms) and ERO results at the Pz electrode (**C**) between active and sham tDCS. * *p* < 0.05.

**Table 1 brainsci-14-00168-t001:** Descriptive (mean and SD) and inferential statistics (degrees of freedom (df), *t* or V, Hedges’ *g*, *p*-value, and adjusted *p*-value) for each behavioral outcome.

		tDCS		df	*t*/V *	*g*	*p*-Value	Adjusted *p*-Value (BH)
		Active	Sham					
Cued Premature Response Task	Premature Responses	27.78 (14.65)	29.75 (15.77)	35	−0.79	0.13	0.432	0.438
	Monetary Gain/Loss	41.82 (66.10)	31.50 (70.73)	37	0.78	0.15	0.438	0.438
	Release time (ms)	240.97 (34.19)	244.89 (42.14)	37	−0.87	0.1	0.388	0.438
Stop-Signal Reaction Time Task	Accuracy Go trials	97.1 (2)	96.2 (3)	30	2.11	0.35	0.043	0.215
	RT Go trials	456.49 (70.32)	443.34 (82.68)	30	1.09	0.17	0.283	0.707
	*p*(respond|signal)	44 (10)	45 (10)	30	−0.38	0.1	0.710	0.71
	SSD	207.71 (55.89)	212.25 (51.45)	30	−0.52	0.08	0.605	0.71
	SSRT	223.77 (46.38)	215.85 (61.63)	30	0.73	0.15	0.471	0.71
Monetary Choice Questionnaire-27	Overall k	0.016 (0.02)	0.015 (0.03)	39	344 *	0.04	0.061	0.12
	Small k	0.033 (0.04)	0.024 (0.04)	39	199 *	0.23	0.004	0.016
	Medium k	0.016 (0.02)	0.015 (0.03)	39	134 *	0.04	0.120	0.12
	Large k	0.012 (0.02)	0.010 (0.03)	39	223 *	0.09	0.106	0.12

* Wilcoxon signed rank test (V).

**Table 2 brainsci-14-00168-t002:** Descriptive (mean and SD) and inferential statistics (degrees of freedom (df), *t* or V, Hedges’ *g*, *p*-value, and adjusted *p*-value) for each EEG outcome.

		tDCS		df	*t*/V *	*g*	*p*-Value	Adjusted *p*-Value (BH)
		Active	Sham					
Target-P3 (250–450 ms)	ERP (µV)	0.99 (5.33)	3.58 (3.14)	32	106 *	0.59	0.001	0.003
	Delta (dB)	2.49 (3.18)	5.01 (6.51)	32	−2.29	0.49	0.015	0.03
	Theta (dB)	3.28 (3.44)	3.77 (4.72)	32	−0.66	0.12	0.805	0.805
Cue-P3 (300–650 ms)	ERP (µV)	0.38 (1.97)	1.39 (1.65)	32	142 *	0.56	0.012	0.036
	Delta (dB)	−0.17 (1.06)	−0.11 (1.04)	32	−0.24	0.03	0.808	0.847
	Theta (dB)	−0.12 (0.91)	−0.07 (1.17)	32	−0.19	0.05	0.847	0.847

* Wilcoxon signed rank test (V).

## Data Availability

Anonymized data may be made available to others upon request and after the approval of a proposal.

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
