# Peer review of "Transcranial Direct Current Stimulation Decreases P3 Amplitude and Inherent Delta Activity during a Waiting Impulsivity Paradigm: Crossover Study"

_brainsci, 2024, doi:10.3390/brainsci14020168_

Round 1

Reviewer 1 Report

Comments and Suggestions for Authors

Mendes et al investigated the effects of tDCS over the rIFG on waiting impulsivity and P3 event-related potential components. A significant decrease in P3 amplitude (cue and target) was reported during active tDCS compared to sham. While these results are potentially interesting, I think the authors should be more transparent about how the EEG signal was preprocessed and how artifacts were handled. To me, it is reasonable that the effects on P3 are merely a result of the way the signal was processed and I think it is important that the authors provide evidence that the observed effects are actually of a physiological nature. 

Introduction

-        Within the introduction several tasks are mentioned followed by how impulsivity or neurophysiological components are affected within the task. However, it seems like the authors assume that the reader knows all these tasks. I feel there is a lack of explanation on the tasks.

Methods/results

-        There seem to be several problems with the correlation analysis as shown in Figure 2. First, neither the x-axis nor y-axis seem to be normally distributed (it seems there are significant outliers). Second, it seems like the correlation is calculated over both sham and active data points. However, for Pearson correlation analysis data points should be independent. Given the present study follows a within design, these data points are not independent. Third, and related to the last point, on a sample of N=40 a correlation of -0.25 would not be significant. It seems like a sample size of N=80 is assumed (but there are only 40 independent data points).

-        On line 227 the authors state that EEG signals were re-referenced, but to what? Another electrode? Common average?

-        Concurrent tDCS-EEG comes with particular artifacts that are less obvious in EEG alone. One is a pronounced cardiac artifact which would be present in the 0.5 – 5 Hz range (Gebodh et al., 2019, https://doi.org/10.1016/j.neuroimage.2018.10.025). Consequently, tDCS-EEG data is anticipated to be a lot noisier. One worry I have is that that the reconstruction of the signal using Artifact Subspace Reconstruction would cause a stronger signal suppression in the active tDCS-EEG signal than in the sham condition. As such, the suppressed P3 results in Figures 3 and 4 might be from artifact cleaning. This seems to become apparent when looking at the earlier peak at around 120 ms, which also seems suppressed. Whether the signals are generally suppressed be verified and the authors should convince the reader that the results are genuine and not simply due to the way that the signal has been processed. For example, the authors should report more details about the artifact removal process using ASR, such as how many artifact components were identified and removed in each condition. It would also be good to report how the results are influenced by using varying artifact detection thresholds.

-        The same point holds for the ERO analysis.

-        There does not seem to be an identifiable cue-P3 peak (Figure 4). Maximum amplitudes of 0.5 and 1.5 are very small compared to generally reported P3 magnitudes. Furthermore, it seems like the differences between active and sham are merely driven by a general drift (sham signal slowly drifting upwards) rather than a specific P3 effect. These results are not very convincing to me.

-        In the topoplot in Figure 4 it seems like the largest amplitude is at the T8 electrode. Is this signal an artifact, or what does it represent?

-        Currently behavioral and electrophysiological results are presented separately. It would also be nice to have an analysis investigating the interaction between both.

Minor

-        When referring to the tDCS electrodes please refer to ‘anode’ and ‘cathode’ instead of ‘active’ and ‘return’.

-        The very first word of the introduction is an acronym. Please start by writing out tDCS in full the first time.

-        I guess the header of 2.2. should not be ‘participants’ (‘procedures’ probably?).

-        In figures 3 and 4 I would anticipate the location of F8 (tDCS electrode) would be more anterior and more lateral.

Comments on the Quality of English Language

English seems good (but I am no native, so I am not a good judge).

Author Response

Dear Reviewer, 

We would like to thank you very much for all the comments and suggestions. We have modified the manuscript according to the comments below, which considerably improved the manuscript.

Please find attached the responses to your comments.

Sincerely,

Sandra 

Reviewer 2 Report

Comments and Suggestions for Authors

1. Could the authors describe the type of study in the title?

2. The authors should revise the manuscript thoroughly regarding extra spaces between the words and paragraphs.

3. The authors should revise the chapters and subchapters. “2.1 Participants; 2.2 Participants.”

4. Methodology, results, and supplementary material are adequate. Please write after the conclusion in the supplementary material chapter that a supplementary material was included.

5. Could the authors highlight the new findings for the literature in their study?

6. Revise the tables and include the meaning of the abbreviations, level of significance, and symbol.

7. The study's limitations should be written in the last paragraph of the discussion.

8. Could the authors provide a figure explaining the design of their study?

9. Could the authors better explain how tDCS was performed? How many sessions? Was there only one, and did it occur during the repetition of the test? How did the authors control the confounding variables from the first control test and those performed after tDCS?

10. Please review the references. Some are missing the year, issue, or page; others are included in the DOI number; others are missing authors.

Author Response

(The authors gave the same response as above.)

Round 2

Reviewer 1 Report

Comments and Suggestions for Authors

I trust the authors addressed all issues to the best of their abilities. There are some doubts from my side concerning potential artifacts of the tDCS on the EEG signal and I would have like to see some mention of this limitation in the discussion section. But besides that I am fine with the changes that have been made.

Author Response

Dear Reviewer,

thank you for your careful revision.

Following your suggestion, we added the following sentence to the manuscript:

"Nevertheless, these studies must enhance their methods in analyzing the EEG data, reducing the impact of artifacts that can interfere with the data. In this particular study, we opted for a previously utilized method, but it is important for future studies to explore alternative approaches to mitigate the potential impact on the data." (lines 514-518)

Sincerely,

Sandra